# Response Variability to Drug Testing in Two Models of Chemically Induced Colitis

**DOI:** 10.3390/ijms24076424

**Published:** 2023-03-29

**Authors:** Roger Suau, Anna Garcia, Carla Bernal, Mariona Llaves, Katharina Schiering, Eva Jou-Ollé, Alex Pertegaz, Arce Garcia-Jaraquemada, Ramon Bartolí, Violeta Lorén, Patri Vergara, Míriam Mañosa, Eugeni Domènech, Josep Manyé

**Affiliations:** 1IBD Research Group, Germans Trias i Pujol Research Institute (IGTP), 08916 Badalona, Spain; rsuau@igtp.cat (R.S.);; 2Centro de Investigación Biomédica en Red Enfermedades Hepáticas y Digestivas (CIBEREHD), 28029 Madrid, Spain; 3Laboratory of Genetic Metabolic Diseases, Faculty of Biosciences, National University of San Marcos, Lima 15088, Peru; 4Institute of Transfusion Medicine and Transplant Engineering, Hannover Medical School, 30625 Hannover, Germany; 5Hepatology Unit IGTP, 08916 Badalona, Spain; 6Department of Physiology, Faculty of Veterinary, Autonomous University of Barcelona, 08193 Bellaterra, Spain; 7Gastroenterology Department, Germans Trias i Pujol University Hospital, 08916 Badalona, Spain

**Keywords:** inflammatory bowel disease, Crohn’s disease, ulcerative colitis, TNBS-induced colitis, DSS-induced colitis, bioluminescence, sex, inflammation

## Abstract

The lack of knowledge regarding the pathogenesis of IBD is a challenge for the development of more effective and safer therapies. Although in vivo preclinical approaches are critical for drug testing, none of the existing models accurately reproduce human IBD. Factors that influence the intra-individual response to drugs have barely been described. With this in mind, our aim was to compare the anti-inflammatory efficacy of a new molecule (MTADV) to that of corticosteroids in TNBS and DSS-induced colitis mice of both sexes in order to clarify further the response mechanism involved and the variability between sexes. The drugs were administered preventively and therapeutically, and real-time bioluminescence was performed for the in vivo time-course colitis monitoring. Morphometric data were also collected, and colonic cytokines and acute plasma phase proteins were analyzed by qRT-PCR and ELISA, respectively—bioluminescence images correlated with inflammatory markers. In the TNBS model, dexamethasone worked better in females, while MTADV improved inflammation in males. In DSS-colitis, both therapies worked similarly. Based on the molecular profiles, interaction networks were constructed to pinpoint the drivers of therapeutic response that were highly dependent on the sex. In conclusion, our results suggest the importance of considering sex in IBD preclinical drug screening.

## 1. Introduction

Crohn’s disease (CD) and ulcerative colitis (UC) are the two most common inflammatory bowel diseases (IBD). Both are chronic inflammatory conditions of the gastrointestinal tract and, similar to many immune-mediated diseases, are characterized by a relapsing-remitting pattern with long-lasting cumulative damage in the involved intestine [1,2]. Their etiology is, at present, poorly understood and relies not only on genetic but also on environmental factors [1,3]. Briefly, unknown environmental factors cause a change in the balance of the gut microbiota that becomes dysbiotic [3,4]. This causes a reaction of the immune system that configures the intestinal barrier for the control of these dysbiotic changes. However, IBD-related genetic mutations induce immune system deficiencies that trigger an aberrant response toward the microbiota, which simultaneously damages the host’s gut [4,5]. The complexity of IBD and limited knowledge about its pathogenesis hamper the development of new therapies that are more effective and precise in the selection of specific targets [6]. Suitable genetic [7], cell transfer [8], and chemically induced colitis [9,10] in vivo models have been developed and widely used to discern the molecular mechanisms at the onset and during IBD [10,11]. However, differences between models and their inability to fully reproduce the pathology as in humans are a barrier to effective drug screening [9].

Among the chemically-induced colitis models, the two main and most widely used in drug testing are the TNBS and DSS experimental models. A single intracolonic administration of the ethanol-dissolved TNBS hapten can elicit significant immunologic responses by rendering colonic proteins immunogenic to the host’s immune system, leading to the development of an excessive Th-1 cell-mediated immune-inflammatory response [12]. TNBS-induced inflammation generates transmural colonic lesions [13] that share the hallmarks of human CD [14]. Severe colitis after TNBS enema is followed by body weight loss, diarrhea, and increased myeloperoxidase (MPO) activity in wild-type mice [10,15,16]. On the other hand, DSS acute colitis is usually induced by continuous administration of 2–7% DSS in drinking water for a short period. DSS administration provokes erosive lesions in the intestinal mucosa with mucin depletion, epithelial degeneration, and necrosis, leading to the disappearance of epithelial cells [10,11,17]. As with TNBS colitis, the clinical manifestations of DSS acute colitis may also include weight loss and diarrhea but more often consist of occult blood in stool, piloerection, and anemia [10,17]. All this leads to epithelial barrier weakening and, therefore, lamina propria transepithelial neutrophilic infiltration, cryptitis, and crypt abscesses [10]. These changes coexist with the loss of components of the tight-junctions such as the zonula occludens complex (ZO-1) [18,19] and a significantly increased production of inflammatory cytokines (TNF-α, IL-1β, IFN-γ, IL-6, and IL-12) in the colon, as well as anti-inflammatory IL-10 [11,19,20,21]. However, the exact mechanisms mediating pathogenesis in both TNBS and DSS models remain elusive. Furthermore, in these IBD models, there are some components, such as sex, which can vary the severity of colitis, even increasing the intra-individual variability of colitis development. In fact, estrogens seem to modulate colitis severity in these models [22,23], and even cytokine expression, such as IL-6, is dependent on the sex of the animal [24]. For instance, DSS colitis was more severe in male mice, and this was estradiol-dependent [23]. There are studies that have found sex-specific responses to drugs in chemically-induced colitis [25,26]. For example, resveratrol, which interacts with estrogen receptors, showed response discrepancies between both sexes in DSS models [25]. In addition, sex-dependent immune features have been found to be unique for each sex in these models [27,28]. Nonetheless, 64% of the preclinical studies with chemically induced colitis models developed so far used only male animals, 21% used females, and only 4% used both sexes. However, IBD incidence in women is somewhat greater than in men [29].

All of this makes drug screening difficult and open to reconsideration. For these reasons, it is necessary to gain knowledge regarding the effect sex has on these chemically-induced models in order to improve the experimental designs for drug screening. For this reason, refined experiments, including the assessment of the inflammatory course based on reactive oxygen species (ROS) detection, should be included in experimental designs. Oxidative stress is defined as an imbalance between oxidants and antioxidants, leading to a disruption of redox signaling and inducing molecular damage [30]. Increased levels of ROS are a hallmark of inflammation and are also elevated in IBD and murine colitis [31,32,33,34]. The main producers of ROS in IBD are activated neutrophils and macrophages. During inflammatory flare-ups, these cells exhibit massive intestinal mucosa infiltration and release large amounts of these species [35]. The analog luminol probe L-012 is able to react with ROS and produce a chemiluminescent signal that can be quantified by bioimaging and has been proven to correlate with intestinal inflammation in murine colitis [32]. This system allows for assessing the inflammatory course of murine colitis in order to reduce the number of animals used and to correct for the variability in the response.

Thus, we aimed to study drug screening variability depending on the model and the mouse sex in two experimental trials, one preventive and the other therapeutic, also considering the inflammatory course based on ROS detection by bioimaging systems.

## 2. Results

### 2.1. Bioluminescence Natural Time-Course in TNBS or DSS Colitis

The bioluminescent images from the TNBS-induced inflammatory course showed a representative profile of ROS production with two inflammatory outbreaks. The first inflammatory outbreak (early) showed high-intensity signals at 48–72 h post-TNBS induction (Figure 1A,B). A second inflammatory outbreak (late) was observed from day five to seven. Although there was slight immunosuppression immediately after TNBS-colitis induction, immunosuppression was much more marked after DSS withdrawal (Figure 2A,B). It was not until two days after DSS (day seven) that an inflammatory flare was detected by ROS-induced bioluminescent signal increase. From this point on, bioluminescence gradually decreased as the course of DSS-colitis progressed.

### 2.2. Bioluminescence Intensity Correlated with Intestinal Immune-Inflammatory mRNA Expression and Plasmatic Acute Phase Proteins

Raw data regarding bioluminescence intensity (BLI) following euthanasia of all the animals were weakly and positively correlated with *Nos2* and *Tnfa* mRNA expression (*p* ≤ 0.05) (Appendix A). Low negative significant correlations were also observed for *Tjp1*, *Tgfb1,* and *Vegfa.* When stratifying by sex, it was observed that all these correlations, except *Tnfa*, were more marked in the male group. In contrast, in females, *Il10*, *Il6,* and *Il1b1* positively correlated with BLI. The correlation of BLI with anti-inflammatory/pro-inflammatory cytokine ratios was calculated to identify the major immunoinflammatory balance in each of the experimental scenarios we studied. All the ratios were weakly correlated with BLI and only significant in female animals (Appendix A), especially *Il10/Il6* and *Il10/Il1b1* ratios.

These correlations were then stratified by the colitis model, trial design, and sex. In the correlations of gene expression with the BLI from the experimental end point of the TNBS therapeutic model, only *Tjp1* showed a moderate negative correlation (Table 1). However, females showed a strong positive significant correlation with *Il1b1*, *Il6,* and *Tnfa*, and with *Il10/Il1b1* and *Tgfb1/Il1b1* ratios (Table 1 and Appendix A). In contrast, in the TNBS-colitis preventive trial, no significant correlation was observed.

Regarding the DSS-colitis therapeutic design (Table 1 and Appendix A), BLI showed a moderate positive correlation with *Il6*, while *Tgfb1* and the *Il10/Il1b1*, *Il10/Il6*, *Tgfb1/Il1b1*, *Tgfb1/Il6,* and *Tgfb1/Tnfa* ratios correlated in the opposite way. However, the strongest correlations were those recorded in males between BLI and *Tgfb1*, *Vegfa,* and *Il10/Tnfa* ratio (Rho ≤ −0.60). In contrast, in female animals, there were no significant correlations at all. Furthermore, in the DSS preventive test, the most noticeable correlations were with *Tjp1* in females (Rho = 0.61) and with *Hif1a* in males (Rho = −0.54), as well as with the *Tgfb1/Il6* ratio.

Regarding the acute phase proteins (APPs), significant correlations with BLI were observed for CRP in the TNBS preventive trial (Rho = −0.42) and for SAA in the DSS therapeutic trial (Rho = 0.64). After separating the groups by sex, there was no significant correlation by model or design (Table 1).

### 2.3. Therapeutic Monitoring Response by Bioluminescence and Macroscopic Evaluations Shows Differences between Colitis Models, Experimental Trials, and Mouse Sex

Colitis monitoring by bioluminescence has made it possible to study the effects of drugs on the time course of colitis. Both treatments attenuated inflammatory activity as captured by ROS-induced bioluminescence in the TNBS therapeutic trial compared to untreated colitic mice (Figure 1A,B). Specifically, MTADV treatment was more efficient than dexamethasone because the reduction in the bioluminescent signal occurred earlier and lasted longer, mainly in male mice, whereas female mice responded better to dexamethasone. This MTADV-induced decrease in the bioluminescent signal in males concurs with an increase in body weight (Figure 3A), whereas the effects of dexamethasone consisted mainly of significantly decreasing the spleen weight, adherences, and the Wallace score (Figure 3B), with no differences between sexes (Appendix A). In contrast, there were no significant BLI changes when the treatments were given preventatively (Figure 1A); but both drugs tended to decrease body weight, especially in females treated with MTADV, as seen in the table in Figure 3A. Additionally, they significantly reduced the number of intestinal strictures (Figure 3B), which were more pronounced in dexamethasone females and MTADV males (Appendix A).

In the DSS therapeutic trial, only dexamethasone-treated females were able to significantly decrease BLI at any time-point in the inflammatory course (Figure 2A,B). Moreover, MTADV worked better in females than males, as shown in the table in Figure 2A. Unexpectedly, MTADV treatment caused an increase in body weight from the start of therapy in this model and design (Figure 4A), and this effect was even more marked in females (see the table in Figure 4A). The remaining macroscopic assessments (Figure 4B) showed results similar to those achieved for dexamethasone in TNBS colitis (Figure 3B), with a significant decrease in intestinal adherences and spleen weight. Furthermore, therapeutic administration of MTADV reduced the Wallace score compared to untreated DSS-colitis mice (Figure 4B), and this was more marked in the female group (Appendix A). Although both treatments given preventively significantly reduced the BLI signal regardless of sex (Figure 2A), only MTADV-treated mice showed resistance to weight loss, as occurs in dexamethasone-treated mice, and this effect is more marked in males (Figure 4A). MTADV treatment also decreased adherences in comparison to the other groups (Figure 4B), mainly in males (Appendix A).

### 2.4. Immune-Inflammatory Signatures in Experimental Colitis Drug Response

Regarding the TNBS therapeutic approach, MTADV increased *Hif1a*, *Il1b1*, *Il6*, *Nos2*, *Tgfb1,* and *Tnfa* expression (Figure 5A). By sex, this increase in expression upon MTADV was more remarkable in males (Appendix A). In the TNBS colitis prevention trial, *Hif1a* and *Tgfb1* expression was elevated by MTADV as in the therapeutic trial, but it also caused an increase in *Tnfa*, *Vegfa,* and *Il10* (Figure 5B). Interestingly, preventive treatment with dexamethasone increased Il1b1 expression. When stratified by sex, all these differences except *Tnfa* were only significant in female animals. Additionally, only females showed an increase in dexamethasone *Nos2* and MTADV *Il1b1* (Appendix A). Comparing sexes, only *Nos2* was increased in male and MTADV-treated controls in comparison to females.

In the case of the therapeutic DSS assay, dexamethasone caused significant changes in decreasing *Nos2* and *Tnfa* expression, as well as an increase in *Vegf1a* mRNA levels, compared to the other groups (Figure 5C). In addition, MTADV decreased *Hif1a* and *Tjp1* in comparison to controls, which was especially evident in females (Appendix A). Moreover, dexamethasone was associated with a higher expression of *Il6* in females, while MTADV decreased the expression of *Il10* in males. In non-treated male mice, *Nos2* and *Il1b1* were more expressed, while female mice showed a higher expression of *Tjp1* and *Vegfa* (Appendix A). In the preventive trial with DSS-colitis, dexamethasone-induced an increase in *Hif1a* and *Il6*, and a decrease in *Tjp1* expression compared to controls, while *Tgfb1* and *Vegfa* expression was increased in the MTADV group, as well as *Nos2* and *Tjp1* in comparison to dexamethasone (Figure 5D). When stratifying animals by sex, the dexamethasone-induced increases, and the MTADV *Nos2* increment occurred mainly in the male group (Appendix A).

CRP and SAA plasma were also compared between treatments, and their level assessments only showed noteworthy results in DSS colitis (Appendix A). In dexamethasone-treated mice, CRP levels were statistically higher than in the other groups in the DSS therapeutic trial. When analyzing these results regarding sex, the difference between the dexamethasone and the control group was only significant in males, while the difference between the dexamethasone and MTADV group was significant in females (Appendix A). Plasma SAA levels were also higher in the prophylactic dexamethasone group (Appendix A). By sex, MTADV-treated male and dexamethasone-treated female mice had lower concentrations of SAA in the TNBS preventive trial and the DSS therapeutic trial, respectively (Appendix A).

### 2.5. Mechanism of Action Based on Co-Expression Interactive Networks

Correlations between study markers were evaluated in order to characterize the prevalent immunoinflammatory mechanism of action in each model of experimental colitis and describe the course of the inflammatory process. In the TNBS therapeutic approach (Table 2, results above the diagonal division in the table are for the therapeutic approach), only significant positive correlations were observed. Among these, *Il1b1* with *Il10* and *Tnfa*; *Il10* with *Nos2* and *Tnfa*, and *Il6* with *Tgfb1* stood out (0.6 < Rho < 0.9). Table 2 shows the remaining, less marked correlations. When stratifying by sex, many of the strongest correlations observed in all the animals remained, but there were other exclusive correlations in each sex group (Appendix A, females, and males, respectively). This was the case of males, which showed a highlighted co-expression of *Tgfb1*, *Tnfa*, *Vegfa,* and *Tjp1* with *Il6*. In the preventive approach, correlations are mostly highly positive (0.6 < Rho < 0.9). *Hif1a* correlates with all the markers, mostly with *Tgfb1* and *Vegfa* (Table 2, results below the diagonal division in the table are for the preventive approach). *Il1b1* also creates a positive network with all the markers except *Il10*, showing a Rho = 0.9 with *Il6*. Similarly, *Tnfa*, *Vegfa,* and *Il6* also show a positive correlation network with many markers. These co-expression profiles were very similar in males and females (Appendix A).

On the other hand, in the DSS therapeutic approach, there were two remarkable correlations: *Tjp1* and *Vegfa* (which, at the same time, were negatively correlated with *Nos2* and *Tnfa*) (Table 3, results above the diagonal division in the table are for the therapeutic approach) and *Nos2* and *Tnfa* (in connection with *Il6* and *Il1b1*). Most correlations also appeared in females and males (Appendix A, respectively). Nevertheless, correlations corresponding to *Hif1a* with *Tgfb1*, *Tnfa,* and *Il1b1*, and the negative correlation of *Nos2* and *Il6* were exclusive to females. In the preventive approach, *Tjp1* had a moderate negative correlation with *Il10*. In contrast, *Tgfb1* and *Il1b1* were highly positively correlated to each other, and to *Hif1a* and *Il6*, and to a lesser extend to *Tnfa*, *Vegfa,* and *Il1b1* (Table 3; results below the diagonal division in the table are for the preventive approach). Most of these significant correlations were only directly present in males after analyzing both sexes separately (Appendix A). However, females showed a high negative exclusive correlation of *Il6* with *Vegfa* (Appendix A).

Regarding the APPs’ correlation to mRNA markers, there were moderate positive correlations between CRP levels and *Il6*, *Hif1a,* and *Il1b1* expression in both trials with DSS-induced colitis (Appendix A). Most of these significant correlations were maintained in females, who also provided additional and stronger correlations between CPR and *Tjp1* and *Nos2* (Rho ≥ |0.84|) (Appendix A), although it should be noted that both sexes *Vegfa* showed a significant correlation with CRP levels. Regarding SAA plasma levels, strong positive correlations with *Il6* and *Il1b1* stand out both in the TNBS-colitis prevention trial and in the DSS-colitis therapeutic trial (Appendix A). These latter correlations were maintained mainly in females (Appendix A), who additionally displayed a strong positive correlation with *Vegfa* and *Tjp1*, and a negative correlation with *Nos2* in therapeutic DSS-colitis and positive ones with *Nos2* and *Tnfa* in preventive TNBS-colitis (Appendix A). Finally, note that the TNBS colitis model also showed strong negative correlations between SAA and *Il6* in females and *Vegfa* or *Nos2* in males.

## 3. Discussion

IBD is a complex disease with multifactorial pathogenesis whose mechanisms are not fully understood as yet. In vivo, experimental models are a useful tool for investigating its pathophysiology and for testing new therapies. To date, it has not been usual to include more than one model of colitis in studies. However, many studies have considered this to take into account different inflammatory phenotypes and colitis courses [11,36,37,38,39,40,41,42]. In this line, this study used two colitis mouse models, TNBS and DSS, considering their already established immune differences [21]. By working on these two different preclinical models with two drug administration strategies (therapeutic and preventive) and sex parity, we built up different immunoinflammatory scenarios linked to the therapeutic response.

In this work, mice with the genetic background C57BL/6 were used to apply a proper chemical colitis inducer dosage. Specifically, this strain shows some features that make the mice more resistant to TNBS-induced colitis but more sensitive to DSS induction [43]. A bioimaging time-course monitoring was used to study the intestinal inflammatory process in a longitudinal view. This method was further validated by relating the BLI to the quantification of inflammatory molecular signatures in both intestinal mucosa and systemic blood. Widely used macroscopic evaluations, a bioluminescent approach, and molecular assessments allowed us to establish the variability between and within the in vivo models in the drug testing of two different treatments, one being dexamethasone, a widely used treatment in IBD, and the other being MTADV, an experimental treatment and an inhibitor of SAA [44].

The follow-up of the colitis time course was performed by capturing ROS production using the L-012 probe in the chemically-induced colitis models. ROS production in the colonic mucosa of experimental colitis models has been correlated with colitis severity [35,45,46]. Real-time in vivo bioluminescence made it possible to quantify the colitis course at each time point and to track intra-individual variability. This meant a reduction in the number of animals per group and a great experimental refinement in monitoring the bowel inflammation time-course of TNBS and DSS colitis models. The inflammatory courses shaped by the BLI of controls matched with what is described in the literature. The bioluminescent images from the TNBS-induced models showed two inflammatory outbreaks. There was a first outbreak probably matching innate immunity cells, mainly neutrophils and macrophages, despite the coexistence of caustic effects of the TNBS enema during this acute phase [47,48]. In the DSS, there was immunosuppression immediately after DSS withdrawal until two days later, when a single inflammatory flare was detected by increasing the intensity of ROS-induced bioluminescence, probably matching an innate immune response [49].

ROS-induced bioluminescence signals positively correlated with the intestinal expression of *Nos2*, although this correlation was more apparent in male animals. *Nos2* encodes for the inducible form of the nitric oxide synthase enzyme, which is responsible for the synthesis of the free radical nitric oxide (NO) from neutrophils and macrophages after microbes or cytokines activate NFκB [50]. NO production has been related to Th1 and Th17 cytokines and has been positively correlated with pro-inflammatory cytokines in plasma in IBD [51,52]. To further refine this validation, we studied the association between BLI and the expression of different markers of oxidative stress and immune inflammation. Thus, specific correlation profiles were identified according to the experimental model and drug administration strategy. When mice were stratified by sex, correlations became more significant, with remarkable anti-inflammatory/pro-inflammatory ratios in female animals since they negatively correlated with BLI. In addition, certain signs of anti-inflammatory activity, mucosal barrier recoveries, and angiogenesis, such as *Tgfb1*, *Vegfa,* and *Tjp1,* were also negatively correlated only in male animals. Altogether, this suggests that the inflammatory processes driving ROS formation and, thus, BLI detection are highly dependent on sex. Specific inflammatory or reparation markers are therefore required to validate this in vivo imaging method (Table 4).

Regarding drug screening, both models showed a BLI decrease associated with both treatments. These therapies induced specific changes in cytokine and oxidative stress markers that depended on the experimental model and sex of the animal. Additionally, most macroscopic morphometric data also matched with a decrease in BLI.

In the TNBS model, the therapeutic administration of dexamethasone produced a decrease in pro-inflammatory cytokines similar to that of IBD, such as *Tnfa*, *Il1b1,* and *Il6*, which matched what was observed in the BLI, contrary to an increase in SAA plasmatic levels. Nevertheless, in the preventive trial, dexamethasone established an inverse situation due to an increase in the relation *Il1b1/Il10*. The situation described above was more obvious in female animals. In contrast, MTADV treatment drove more changes in the male immune response. Here, there was an unexpected increment in pro-inflammatory cytokines and oxidation markers in both trials (*Hif1a* and *Tnfa*). Nevertheless, the treatment decreased the BLI in the therapeutic approach and increased reparation markers (*Vegfa* and *Tgfb1*) in the preventive design. In this trial, there was also a decrease in plasmatic levels of SAA, as expected due to the targeting action of MTADV [44]. Similarly, in another study, MTADV showed contradictory results by reducing some inflammatory markers, such as SAA, while it increased IFN-γ in TNBS mice [44]. Moreover, when looking at the correlations between mRNAs expressions and BLI, the above-mentioned *Il1b1*, *Il6,* and *Tnfa* showed a significant correlation (Rho ≥ 0.60) with BLI, mainly in females, regardless of treatment.

In the DSS model, BLI was reduced in the drug-treated groups and more pronounced in prevention and exclusive to females in the therapeutic trial. In addition, dexamethasone produced a reduction in *Nos2,* mainly in males, as well as a *Vegfa* in the therapeutic trial. In fact, the pro-angiogenic VEGFA pathway was related to the mechanism of action within glucocorticoid response in a UC cohort [53]. Furthermore, a decrease in *Tnfa* following the same trend as the BLI decrement was also seen in females. Both treatment strategies in DSS-colitis showed a loss in weight regardless of sex, which is contradictory to our understanding that glucocorticoids induce weight gain. Nevertheless, this fact is dose and time-dependent [54,55], and our experiment does not imply chronic exposure to this treatment. On the other hand, treatment with MTADV induced an increment in the expression of *Nos2* regardless of sex, while, in males, *Tnfa,* among others, was increased, showing a pro-inflammatory profile. This matched with a decrease in the BLI in females but not in males in the therapeutic approach. Nevertheless, males showed strong negative correlations linking bioluminescent signals with a decrease in reparation and healing processes (rho = −0.60 and −0.81 for *Tgfb1* and *Vegfa*, respectively), whose indicators are the most affected by treatments.

The dexamethasone results show great differences between the sexes. It is known that there are sex-dependent differences in response to glucocorticoids. Recently, a study observed an overrepresentation of transcriptomic targets of the glucocorticoid receptor pathway in DSS colitis [27]. It is known that females exert more potent pro-inflammatory processes due to significantly higher estrogen liberation and a lower presence of androgens than males, these being pro-inflammatory and anti-inflammatory, respectively [56]. In line with this, we found higher *Tnfa* levels in females in the TNBS-colitis model compared to male controls in the therapeutic approach. In fact, glucocorticoids can inhibit the action of these hormones, suggesting that sex is an important point to check whenever glucocorticoids are used as treatment [56]. There is also a general increase in *Il6* [57,58] in DSS colitis in male animals in the therapeutic trial. Actually, the role of IL-6 has been reported to be sex-dependent in many diseases and situations [24].

Comparing both models, we found that females responded better to dexamethasone in TNBS than in the DSS model. As for males, an inverse situation is observed. MTADV treatment worked better in males with TNBS-induced colitis and females with DSS-induced colitis, highlighting the potential importance of using different animal models and both sexes in drug screening studies. In addition, while poor results were obtained in the preventive trial with TNBS, many inflammatory clinical and molecular markers were decreased in the DSS model. These results rely on the difference in the pattern of administration of the inductor of colitis. The fact that the preventive treatments started while DSS was being given to the animal may explain why the preventive treatment worked. Although the inflammatory process was ongoing in these stages, the drivers of this process may have been inhibited, and further inflammation was precluded. Moreover, in the TNBS preventive trial, we did not observe many changes in cytokines with the treatments nor a correlation between these markers and BLI. The TNBS acute phase might be confounded with a caustic reaction to the chemical itself [47,48]. The increase in BLI might be a consequence of the production of ROS from the necrotizing effect of TNBS enema on intestinal epithelial cells.

In the TNBS colitis model, *Tnfa*, *Il1b1,* and *Hif1a*, the latter predominantly in the preventive trials, built interesting correlation networks, thereby highlighting the fact that they are the cytokines that drive the inflammatory process in female animals. In males, *Tnfa* and *Il6* were central cytokines, networking with other cytokines regardless of the treatment. In fact, *Il6* correlated with some T-regulatory cells (Treg), angiogenesis, and reparation processes. Moreover, regardless of the animal sex, the inflammatory *Tnfa*, *Il6,* and *Il1b1* correlated positively with BLI, as well as with systemic SAA. Indeed, this is consistent with the fact that these inflammatory cytokines are potent inducers of APPs in the liver [59]. Hence, *Tnfa* becomes a central cytokine in the TNBS model, as also happens in CD [60], regardless of the sex of the animal. This was also shown in the preventive approach. In fact, *Tnfa*, *Il1b1,* and *Il6* play a central role in TNBS colitis due to T cell, neutrophil, and macrophage infiltration [21], as well as in T CD4+ differentiation to Th1 and Th17 cells as shown in Figure 6. Unfortunately, *Il4*, *Il17a,* and *Ifng* expressions were too low in our samples and precluded us from defining, more precisely, the immune profile of the intestinal mucosa. There is also prominent participation of *Il10* and *Tgfb1* in the immune networks of TNBS-colitis regardless of the trial and sex that counteracts the inflammatory imbalance [21]. They induce naïve T lymphocytes to become Tregs (as shown in Figure 6). Our results match those obtained in the same model by Silva et al. 2022 [61], and IL-10 therapy ameliorates TNBS colitis [62]. Moreover, *Vegfa* was also correlated positively to *Il10* and *Tgfb1* networks and negatively to SAA, the latter in the therapeutic approach. *Vegfa* has been related to inflammation, angiogenesis, and leucocyte rolling in IBD [63]. Finally, *Hif1a* also built interesting co-expression networks in the preventive approach linked to acute inflammation. In both human IBD [64] and experimental colitis [65,66], hypoxia-induced HIF1A is critical for many mucosal barrier-protective genes and angiogenesis [64].

In DSS-induced intestinal inflammation, mucosal *Il6* and plasmatic SAA correlated positively with BLI, while *Tgfb1* correlated negatively in the therapeutic approach. On the other hand, both BLI and *Nos2* correlated negatively with reparative *Tjp1* and *Vegfa* expression, mainly in females. In contrast, in the preventive trial, cytokine correlations were similar to those in the TNBS and mainly influenced by males. Here, there was also a negative correlation between *Il6* and BLI, suggesting a certain immunosuppressive effect in this acute inflammatory stage. This contrasts with the therapeutic model, in which reparatory processes were the protagonists. DSS-induced barrier weakness causes bacterial translocation, which activates functions driven by toll-like receptors in both epithelial cells and neutrophils [21]. Among these is the synthesis of NFkB-induced cytokines, but also an expression of epithelial barrier genes and increasing endothelial permeability and adherence for neutrophil infiltration (Figure 6). Moreover, at the moment the mice were euthanized in the preventive trials, the APPs correlated with the pro-inflammatory markers that are their inducers [59]. When *Tnfa* is increased, as in the DSS model, this factor reduces the expression of tight junction proteins, as already seen in IBD [67,68], and may also affect angiogenesis. A common feature in both approaches is *Hif1a* expression, which, as mentioned above, is related to IBD and DSS-induced colitis [69,70]. There were many important differences in sexes here, too. In fact, in the therapeutic approach, male controls showed more pro-inflammatory signals than females, whereas females showed more *Tjp1* and *Vegfa*. In addition, DSS causes a loss of reparatory response upon inflammation [17], but this may happen mainly in male mice since they are more susceptible to DSS [10], whereas females exert a higher reparatory response.

## 4. Materials and Methods

### 4.1. Mice

Ten- to twelve-week-old C57BL/6J mice (male and female) were purchased from Jackson Laboratory (Charles River; Barcelona, Spain). The mice were housed in the Germans Trias i Pujol Research Institute (IGTP) conventional facility, and the studies were conducted in compliance with animal research guidelines. All the experiments were approved by the Institutional Animal Care and Use Committee of the IGTP and authorized by the Catalan Government.

### 4.2. Chemically Induced IBD Models

The study was performed in two IBD models using TNBS and DSS as colitis inducers. For the TNBS colitis, mice were anesthetized using isoflurane, and then each mouse received an intrarectal enema of 3 mg TNBS dissolved in 100 mL of ethanol/H2O (50%/50%, *v/v*). In the case of DSS-induced colitis, mice received 2.5% DSS in drinking water for five consecutive days. Mice were euthanized at the specific times established according to the previous tests performed. The endpoints for the therapeutic trials of TNBS and DSS were on days seven and ten, respectively, and the endpoints for the preventive trials of TNBS and DSS were set on days three and seven, respectively.

### 4.3. Experimental Groups

In both models of experimental colitis, mice were randomly divided into four groups that remained constant along the different experimental designs: non-treated sham colitis group (healthy); TNBS/DSS colitis + vehicle (PBS, s.c); TNBS/DSS colitis + dexamethasone (1.5 mg * kg^−1^, s.c.); and TNBS/DSS colitis + MTADV (3 mg * kg^−1^, s.c.); Drug doses were chosen according to bibliographic criteria and previous dose-response screenings in both experimental models. Each group included ten animals (half of which were females).

In the therapeutic assay, PBS, dexamethasone, or MTADV were administered by subcutaneous injection from day four after intrarectal instillation of TNBS and were continued until day seven (Figure 7A). In vivo real-time bioluminescence was performed from 48 h to day seven of the TNBS course. To evaluate the potential preventive effect of the drugs in the TNBS group, mice received PBS or each drug from three days from −48 h to day zero before TNBS instillation (Figure 7B). Bioimaging was performed at baseline (before TNBS administration) and at 24, 48, and 72 h after TNBS or sham colitis induction.

In the therapeutic approach, the treatments were distributed in four subcutaneous doses of PBS, dexamethasone, or MTADV dispensed between days seven and ten (Figure 7C). Bioimaging was performed at the pre-colitis stage (day −1) and from day five to day ten. In the preventive DSS-colitis trial, mice received three drug doses, during the last three days of DSS administration, from day three to day five (Figure 7D). Bioimaging was performed at the pre-colitis stage (day −1) and from day five to day seven.

### 4.4. Real-Time In Vivo ROS Quantification by Bioluminescent Imaging

Mice were anesthetized by isoflurane and injected with probe L-012 at 25 mg * kg^−1^ (i.v.) via the retro-orbital sinus, which produces a chemiluminescent signal when reacting to ROS [32]. Image acquisition was performed with the IVIS Lumina II imaging system (PerkinElmer Imaging Systems, Waltham, MA, USA). Region of interest (ROI) was measured using Living Image software 4.7.2 and expressed as average radiance (photons/second/cm^2^/sr). A baseline image was taken before colitis induction (background), and the colitis time course was followed, capturing images at previously selected time points (Figure 7A–7D). The bioluminescence images were normalized using the baseline values of each mouse as background and expressing their values as FC. In the therapeutic trials, another FC calculation was added with the mean values of the days between the start of the colitis and the treatment to evaluate the treatment effect.

### 4.5. Sampling and Macroscopic Colitis Evaluation

Animals were weighed, and their well-being was assessed daily during the experimental time course. After euthanasia, plasma samples were obtained, and the colon and spleen were dissected and weighed. The colonic weight/length ratio, the number of colonic strictures, and hyperemia were measured. These macroscopic features were grouped and evaluated together according to the Wallace score (Appendix A) [71]. RNAlater-preserved colonic tissue and plasma samples were kept frozen at −80°.

### 4.6. Quantitative Real-Time Polymerase Chain Reaction

Colon samples in RNA were later thawed and washed with PBS, and fragmented using a mortar cooled in liquid nitrogen. Then, 25 mg of frozen tissue was homogenized in 700 ul of QIAzol reagent (QIAgen, Madrid, Spain) using a gentleMACS™ tissue dissociator (Miltenyi Biotech, Teterow, Germany). After chloroform was added, the sample was separated into three phases. The upper phase was then used for RNA purification using the *miRNeasy* Mini kit in the QIAcube system (QIAGEN, Hilden, Germany) following the manufacturer’s protocol. RNA integrity was evaluated by an electrophoresis system Agilent 6000 Nano kit (Agilent Technologies, Santa Clara, CA, USA) using RNA Bioanalyzer chips. Only RNA integrity numbers equal to or higher than 6.5 were deemed good and processed. A total of 1 μg of total RNA was retrotranscribed to cDNA using the PrimeScriptTM RT reagent (Perfect Real Time kit; Takara, Shiga, Japan) following the manufacturer’s instructions. TaqMan^®^Assays (ThermoFisher Scientific, Madrid, Spain) for qRT-PCR of *Il6*, *Il10*, *Tgfb1*, *Tnfa*, *Vegfa*, *Hif1a*, *Nos2*, *Il1b1* and *Tjp1* (TaqMan probes: Mm00446190_m1, Mm00439614_m1, Mm01178820_m1, Mm00443260_g1, Mm00437306_m1, Mm00468869_m1, Mm00440502_m1, Mm00434228_m1, and Mm01320638_m1, respectively) were used. PCR thermal cycling included initial denaturing at 95 °C for 10 s, 40 cycles of 95 °C for 15 s, and 60 °C for one minute (LightCycler480 system; Roche Diagnostics, Basel, Switzerland). Data from qRT-PCR were calculated using the 2^−ΔΔCt^ method [72]. The results were normalized with both *Gapdh* and *B2m* (Mm99999915_g1 and Mm00437762_m1, respectively) housekeeping genes.

### 4.7. Enzyme-Linked Immunosorbent Assay

Plasma samples were thawed, vortexed, and centrifuged for 10 min at 1000 rpm. C-Reactive Protein (CRP) plasma levels were measured using the Mouse CRP ELISA Kit (Invitrogen by Thermo Fisher Scientific, Waltham, MA, USA) according to the manufacturer’s instructions, diluting all samples 2000-fold. Serum amyloid antigen (SAA) plasmatic concentrations were determined using the Phase SAA Murine Assay Kit (Tridelta Development Ltd., Maynooth, Ireland), diluting the samples between 50–100-fold for the TNBS-colitis and 500–8000-fold for DSS-colitis. All samples were analyzed in duplicates, and OD was measured in the Varioskan microplate reader (Thermo Fisher Scientific, Waltham, MA, USA). Standard curves were generated. For the CRP quantification, a four-parameter algorithm in the MyCurveFit Add-in in Microsoft Excel 2016 software (version R2 v1.0102.823, MyAssays Ltd, Brighton, United Kingdom).

### 4.8. Statistics

All quantitative data were checked for normality using a Shapiro-Wilk test. We also applied the Mann–Whitney U test in pairwise comparisons between different animal groups at the same time point. The comparisons of the qRT-PCR and ELISA data between treatment groups and sexes were performed with the Kruskal–Wallis test with a post hoc Dunn’s test analysis or a Mann–Whitney U test. Comparisons with *p* ≤ 0.05 were significant statistically. Correlation analyses were performed using Spearman’s test. Correlations with 0.20 ≤ Rho < 0.40 were considered weak, those with 0.40 ≤ Rho < 0.60 moderate, and those with Rho ≤ 0.60 strong. All these analyses were performed with R software version 4.2.0.

## 5. Conclusions

Preventive MTADV induces reparatory activity in both models, even though it induces inflammation when used as treatment. However, dexamethasone plays a more remarkable role in therapy, mainly in the DSS model, by decreasing inflammatory factors. Nevertheless, these results are highly dependent on the sex of the animal. We, therefore, conclude that the use of the TNBS and DSS experimental colitis models is a suitable strategy for evaluating the pathology of IBD and drug screening. However, features such as the variability between individuals of each group and the sex of the animal should be considered and can be solved by using the BLI or including both sexes in the groups. In addition, the immunity of each type of model and the course of the inflammation must be taken into account depending on the molecular focus of the study.

## Figures and Tables

**Figure 1 ijms-24-06424-f001:**
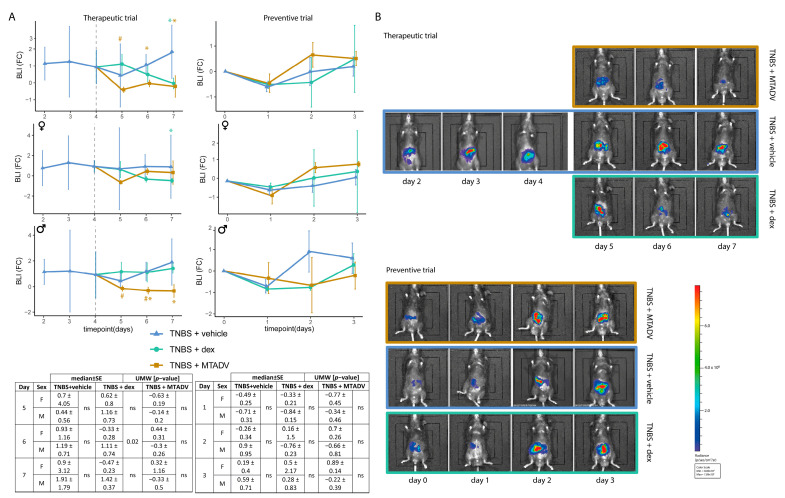
Bioluminescence evaluations in the follow-up of the TNBS model. (**A**) BLI is represented as median FC ± SE in the follow-up, represents therapeutic and preventive approaches, and is stratified by sex. Colors and dots denote treatment groups. Tables showing the statistical test (UMW) between sexes at each time point are given. (**B**) Graphical representation of bioluminescent images of each trial and treatment at each time point. Blue color represents the minimum pixel level, and red color the maximum pixel density. Asterisks indicate differences between treatments and vehicle administration, whereas pads mark differences between dexamethasone and MTADV treatments at each time point (Kruskal–Wallis with posthoc Dunn’s test). Significance levels: */^#^ 0.01 ≤ *p*-value ≤ 0.05. BLI, bioluminescence intensity; dex, dexamethasone; F, female; FC, fold-change; M, male; MTADV, 5-MER peptide methionine-threonine-alanine-aspartic acid-valine; SE, standard error; TNBS, trinitrobenzenesulfonic acid; UMW, U Mann–Whitney test.

**Figure 2 ijms-24-06424-f002:**
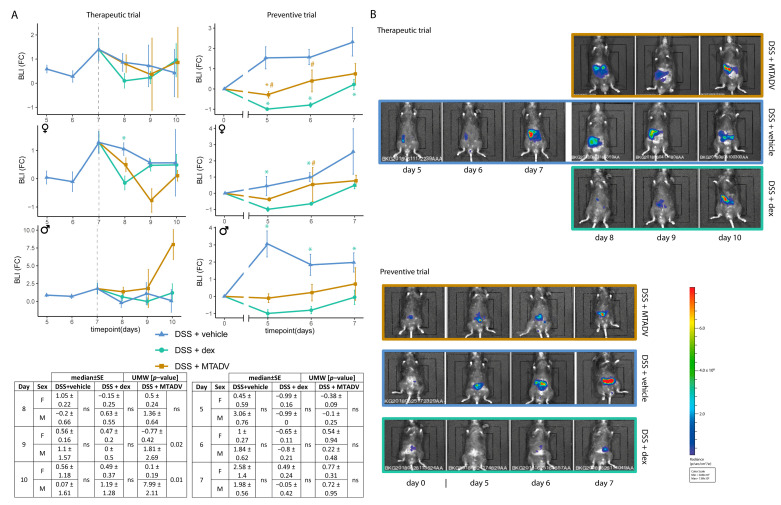
Bioluminescence evaluations in the follow-up of the DSS model. (**A**) BLI is represented as median FC ± SE in the follow-up, represents therapeutic and preventive approaches, and is stratified by sex. Colors and dots denote treatment groups. Tables showing the statistical test (UMW) between sexes at each time point are given. (**B**) Graphical representation of bioluminescent images of each trial and treatment at each time point. Blue color represents the minimum pixel level, and red color the maximum pixel density. Asterisks indicate differences between treatments and vehicle administration, whereas pads mark differences between dexamethasone and MTADV treatments at each time point (Kruskal–Wallis with posthoc Dunn’s test). Significance levels: */^#^ 0.01 ≤ *p*-value ≤ 0.05. BLI, bioluminescence intensity; dex, dexamethasone; DSS, dextran sulfate sodium; F, female; FC, fold-change; M, male; MTADV, 5-MER peptide methionine-threonine-alanine-aspartic acid-valine; SE, standard error; UMW, U Mann–Whitney test.

**Figure 3 ijms-24-06424-f003:**
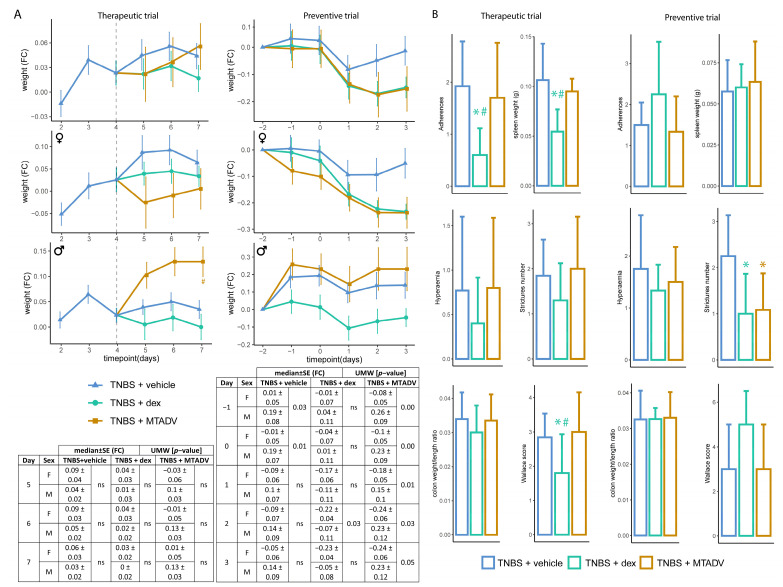
Macroscopic evaluations in the follow-up of the TNBS model. (**A**) Body weight is represented as median FC ± SE in the follow-up, represents therapeutic and preventive approaches, and is stratified by sex. Colors and dots denote treatment groups. Tables showing the statistical test (UMW) between sexes at each time point are given. (**B**) Morphometric data are represented for both therapeutic and preventive trials. Data are represented as bar plots (median ± SD). Each variable and metric are given on the Y-axis, while colors indicate the treatment group. Asterisks indicate differences between treatments and vehicle administration, whereas pads mark differences between dexamethasone and MTADV treatments at each time point (Kruskal–Wallis with posthoc Dunn’s test). Significance levels: */^#^ 0.01 ≤ *p*-value ≤ 0.05. BLI, bioluminescence intensity; dex, dexamethasone; F, female; FC, fold-change; M, male; MTADV, 5-MER peptide methionine-threonine-alanine-aspartic acid-valine; SD, standard deviation; SE, standard error; TNBS, trinitrobenzenesulfonic acid; UMW, U Mann–Whitney test.

**Figure 4 ijms-24-06424-f004:**
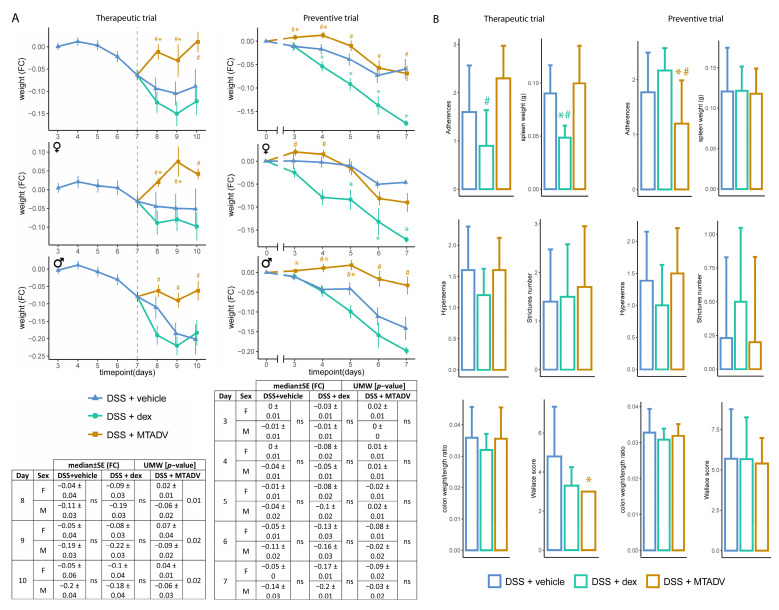
Macroscopic evaluations in the follow-up of the DSS model. (**A**) Body weight is represented as median FC ± SE in the follow-up, represents therapeutic and preventive approaches, and is stratified by sex. Colors and dots denote treatment groups. Tables showing the statistical test (UMW) between sexes at each time point are given. (**B**) Morphometric data are represented for both therapeutic and preventive trials. Data are represented as bar plots (median ± SD). Each variable and metric are given on the Y-axis, while colors indicate the treatment group. Asterisks indicate differences between treatments and vehicle administration, whereas pads mark differences between dexamethasone and MTADV treatments at each time point (Kruskal–Wallis with posthoc Dunn’s test). Significance levels: */^#^ 0.01 ≤ *p*-value ≤ 0.05. BLI, bioluminescence intensity; dex, dexamethasone; DSS, dextran sulfate sodium; F, female; FC, fold-change; M, male; MTADV, 5-MER peptide methionine-threonine-alanine-aspartic acid-valine; SD, standard deviarion; SE, standard error; UMW, U Mann–Whitney test.

**Figure 5 ijms-24-06424-f005:**
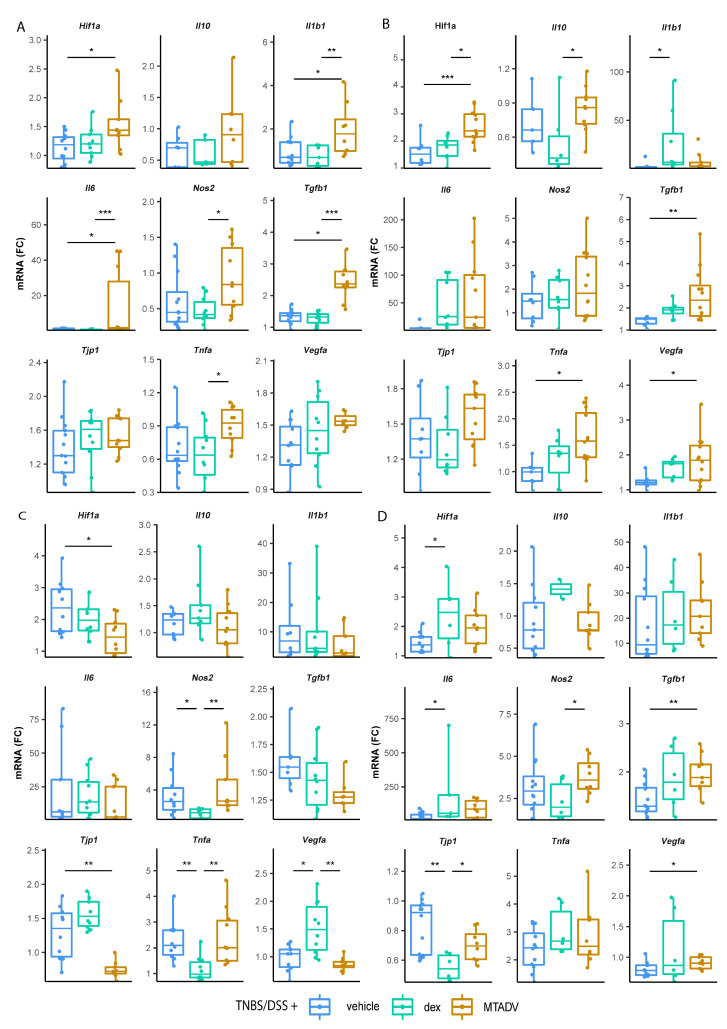
mRNA expression of inflammatory markers in each treatment group in the affected mucosa. qRT-PCR quantified mRNA expression as FC of inflammatory markers for both therapeutic and preventive trials from the TNBS ((**A**) and (**B**), respectively)) and DSS models ((**C**) and (**D**), respectively). Data are represented as points summarized in boxplots, indicating IQR, and bars, indicating maximum and minimum. The median point is depicted as a bar inside the box. Treatment groups are differentiated using colors. Asterisks mark significance between the groups indicated by lines (Kruskall–Wallis test with a posthoc Dunn’s test). Significance levels are: * *p*-value ≤ 0.05; ** *p*-value ≤ 0.01; *** *p*-value ≤ 0.001. Dex, dexamethasone; DSS, dextran sulphate sodium; FC, fold-change; IQR, interquartile range; MTADV, 5-MER peptide methionine-threonine-alanine-aspartic acid-valine; TNBS, trinitrobenzenesulfonic acid.

**Figure 6 ijms-24-06424-f006:**
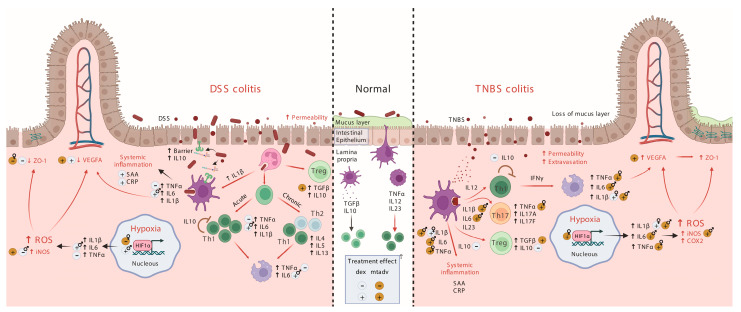
Representation of the inflammatory process induced in the colonic mucosa of both TNBS and DSS models. TNBS changes the conformation of antigens in the colonic mucosa, making it alien to the host immune system and thereby initiating transmural colitis. Dendritic cells recognizing these antigens drive the inflammatory process to Th1 and Th17 phenotypes which are also characterized by infiltration of the lamina propria with CD4+ T cells, neutrophils, and macrophages by the systemic production of APPs. It also drives T-cell differentiation to the T-reg phenotype but shows a more pronounced increment of pro-inflammatory cells. The hypoxic environment and macrophages activated by Th1 cells promote angiogenesis and mucosal immune barrier reparation (tight junctions) [21]. DSS causes disruption of the intestinal epithelial barrier and, thereby, the entry of luminal bacteria or bacterial antigens into the mucosa. Neutrophil and dendritic cell detection of PAMPs by TLR, among others, provoke the start of inflammation, characterized by a Th1 phenotype in acute models and a mix of both Th1 and Th2 in chronic colitis. Altogether, this increases the production of APPs and decreases angiogenesis and mucins and tight junction protein expression with an imbalance between pro-inflammatory and anti-inflammatory cytokines [21]. Created with BioRender.com. APP, acute phase protein; PAMP, pathogen-associated molecular pattern; T helper cells; TLR, toll-like receptor; TNBS, trinitrobenzenesulfonic acid, T-reg, T regulatory cells.

**Figure 7 ijms-24-06424-f007:**
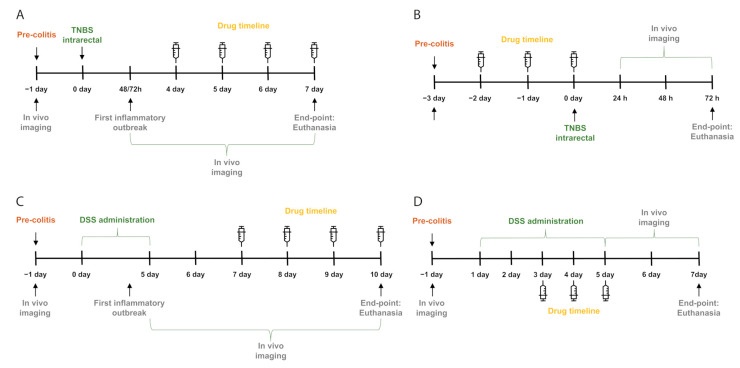
Graphical representation of the experimental design of the four chemically induced colitis approaches. Experimental design throughout time for the preventive (**A**) and therapeutic (**B**) approaches for TNBS and the preventive (**C**) and therapeutic (**D**) trials in DSS mice. Time is represented as days. Negative days are prior to colitis inducer administration. Colitis inducer administration and in vivo imaging are indicated as arrows or brackets. Drug delivery is marked as syringes. End-point refers to the last day of the trial.

**Table 1 ijms-24-06424-t001:** BLI raw data correlation to mRNA and protein levels from inflammatory and oxidative markers of the animals stratified by colitis model, experimental trial, and sex.

				Intestinal mRNA									Plasmatic APP	
				*Il10*	*Tgfb1*	*Il6*	*Vegfa*	*Tjp1*	*Nos2*	*Tnfa*	*Hif1a*	*Il1b1*	CRP	SAA
**Both** **sexes**	**TNBS**	**Ther**	*p*	0.77	0.56	0.41	0.82	**0.02**	0.85	0.23	0.38	0.69	0.74	0.63
			R	0.07	−0.11	0.16	−0.04	**−0.42**	0.03	0.22	−0.16	0.08	0.06	0.12
		**Prev**	*p*	0.46	0.77	0.48	0.85	0.96	0.08	0.91	0.78	0.61	**0.03**	0.32
			R	−0.17	0.06	−0.15	−0.04	0.01	0.33	0.02	0.06	−0.11	**−0.42**	−0.26
	**DSS**	**Ther**	*p*	0.09	**0.01**	**0.04**	0.13	0.17	0.61	0.12	0.38	0.08	0.12	**0.01**
			R	−0.32	**−0.46**	**0.40**	−0.30	−0.27	−0.10	0.30	0.17	0.35	0.30	**0.64**
		**Prev**	*p*	0.50	0.35	**0.05**	0.53	0.10	0.10	0.58	0.13	0.69	0.66	0.39
			R	−0.15	−0.19	**−0.39**	−0.13	0.32	0.32	−0.11	−0.30	−0.08	−0.09	−0.23
**F**	**TNBS**	**Ther**	*p*	0.84	0.42	**0.01**	0.90	0.06	0.67	**0.01**	0.91	**0.02**	0.64	0.52
			R	−0.08	0.22	**0.61**	−0.04	−0.48	0.11	**0.64**	0.03	**0.60**	0.13	0.25
		**Prev**	*p*	0.82	0.42	0.17	0.47	0.96	0.36	0.96	0.95	0.31	0.13	0.64
			R	0.08	−0.22	−0.37	−0.20	−0.01	−0.25	0.01	−0.02	−0.28	−0.41	−0.17
	**DSS**	**Ther**	*p*	0.68	0.08	0.28	0.22	0.21	0.75	0.83	0.06	0.34	0.22	0.58
			R	0.12	0.47	0.30	0.34	0.34	−0.09	−0.06	0.50	0.27	0.34	0.22
		**Prev**	*p*	0.31	0.53	0.13	0.23	**0.04**	0.48	0.07	0.72	0.53	0.95	0.44
			R	−0.34	−0.20	−0.47	0.38	**0.61**	0.22	0.55	−0.12	0.20	0.02	−0.30
**M**	**TNBS**	**Ther**	*p*	0.79	0.07	0.14	0.81	0.20	0.93	0.39	0.18	0.10	0.65	0.71
			R	0.10	−0.45	−0.45	−0.06	−0.33	−0.02	−0.22	−0.34	−0.43	0.12	0.13
		**Prev**	*p*	0.97	0.61	0.97	0.88	0.39	0.11	0.45	0.61	0.82	0.82	0.56
			R	0.02	0.17	−0.02	0.05	−0.29	0.46	0.25	0.17	0.08	−0.08	−0.29
	**DSS**	**Ther**	*p*	0.18	**0.00**	0.88	**0.03**	**0.05**	0.26	0.22	0.58	0.16	0.30	0.92
			R	−0.40	**−0.81**	0.05	**−0.60**	**−0.57**	0.37	0.37	−0.17	−0.43	−0.31	−0.09
		**Prev**	*p*	0.67	0.58	0.08	0.17	0.33	0.23	0.19	**0.04**	0.36	0.37	0.30
			R	−0.14	−0.15	−0.46	−0.38	0.27	0.33	−0.36	**−0.54**	−0.25	−0.26	−0.37

Spearman correlation analyses. Red and blue colors indicate significant (*p* ≤ 0.05) negative and positive correlations. APP, acute phase proteins, BLI, Bioluminescence intensity; CRP, C-reactive protein; DSS, dextran sulfate sodium; F, female; M, male; *p*, *p*-value; Prev, Preventive; R, Rho; SAA, serum amyloid antigen; Ther, Therapeutic; TNBS, trinitrobenzenesulfonic acid.

**Table 2 ijms-24-06424-t002:** Correlation between inflammatory and oxidative stress mRNA expression in the TNBS model in both therapeutic and preventive approaches.

	*Hif1a*	*Il10*	*Il6*	*Nos2*	*Tgfb1*	*Tnfa*	*Vegfa*	*Tjp1*	*Il1b1*	
** *Hif1a* **		0.17	**0.43**	0.09	**0.58**	0.08	**0.59**	**0.52**	0.11	**Ther**
** *Il10* **	**0.49**		0.08	**0.61**	0.23	**0.66**	−0.15	−0.09	**0.82**	
** *Il6* **	**0.70**	0.28		0.29	**0.61**	**0.50**	**0.52**	0.33	**0.52**	
** *Nos2* **	**0.41**	**0.55**	0.27		0.17	**0.57**	0.05	−0.15	**0.47**	
** *Tgfb1* **	**0.75**	0.23	**0.76**	0.32		**0.40**	0.28	**0.35**	**0.39**	
** *Tnfa* **	**0.64**	0.38	**0.69**	**0.61**	**0.62**		−0.21	−0.26	**0.76**	
** *Vegfa* **	**0.75**	0.23	**0.87**	0.18	**0.71**	**0.61**		**0.48**	−0.28	
** *Tjp1* **	**0.40**	0.36	0.17	−0.05	0.29	−0.01	0.31		−0.24	
** *Il1b1* **	**0.61**	0.05	**0.90**	**0.46**	**0.72**	**0.66**	**0.79**	−0.05		
	**Preventive**									

Spearman linear regression analyses. Red and blue colors indicate significant (*p* ≤ 0.05) negative and positive correlations, respectively. Grey diagonal separates therapeutic (top) from preventive (bottom) correlations. Prev, Preventive; Ther, Therapeutic; TNBS, trinitrobenzenesulfonic acid.

**Table 3 ijms-24-06424-t003:** Correlation between inflammatory and oxidative stress mRNA expression in the DSS model in both therapeutic and preventive approaches.

	*Hif1a*	*Il10*	*Il6*	*Nos2*	*Tgfb1*	*Tnfa*	*Vegfa*	*Tjp1*	*Il1b1*	
** *Hif1a* **		0.18	0.32	−0.13	0.30	0.14	0.27	**0.46**	**0.44**	**Ther**
** *Il10* **	−0.13		0.1	−0.19	**0.5**	−0.23	**0.43**	0.34	0.20	
** *Il6* **	**0.74**	0.23		**0.4**	−0.03	**0.38**	−0.22	−0.19	**0.91**	
** *Nos2* **	−0.02	−0.37	−0.13		−0.28	**0.82**	**−0.68**	**−0.73**	**0.39**	
** *Tgfb1* **	**0.73**	0.21	**0.76**	−0.16		0.05	**0.59**	**0.52**	0.09	
** *Tnfa* **	**0.49**	−0.16	**0.44**	**0.44**	**0.39**		**−0.43**	**−0.43**	0.36	
** *Vegfa* **	0.26	−0.09	0.20	−0.24	**0.58**	0.34		**0.82**	−0.19	
** *Tjp1* **	0.08	**−0.58**	−0.19	0.05	−0.11	0.11	0.16		−0.13	
** *Il1b1* **	**0.64**	0.17	**0.77**	0.08	**0.68**	**0.57**	0.21	−0.10		
	**Prev**									

Spearman linear regression analyses. Red and blue colors indicate significant (*p* ≤ 0.05) negative and positive correlations, respectively. Grey diagonal separates therapeutic (top) from preventive (bottom) correlations. DSS, dextran sulfate sodium; Prev, Preventive; Ther, Therapeutic.

**Table 4 ijms-24-06424-t004:** Proposed remarkable mRNA mucosal and plasmatic APPs inflammatory and reparation markers with a moderate-high correlation with ROS detection using bioluminescence stratified by model, approach, and sex.

Model	Approach	Both Sexes	F	*M*
**TNBS-colitis**	Therapeutic	*Tjp1*	*Tnfa*	*Tgfb1^t^*
			*Il6*	
			*Il1b1*	*Il6^t^*
			*Tjp1^t^ Il10/Il1b1 Tgfb1/Il1b1 Il10/Il6^t^*	*Il1b1^t^ Il10/Il6^t^*
	Preventive	CRP	CRP^t^	*Nos2^t^*
			*Il10/Il6^t^*	
			*Il10/Il1b1^t^*	
**DSS-colitis**	Therapeutic	*Tgfb1*	*Tgfb1^t^*	*Tgfb1*
		*Il6*		*Il6*
		*SAA*		*Vegfa*
		*Il10/Il6*		
		*Tgfb1/Il6*	*Hif1a^t^*	*Tjp1*
		*Tgfb1/Tnfa*		*Il10/Tnfa*
		*Il10/Il1b1*		*Tgfb1/Tnfa*
		*Tgfb1/Il1b1*		
	Preventive	*Il6*	*Tjp1*	*Hif1a*
			*Tnfa^t^*	*Il6^t^*
		*Tgfb1/Il6*	*Tgfb1/Il6^t^*	*Tgfb1/Il6*

^t^ correlation was moderate but tendential. APP, Acute phase protein; CRP, C-reactive protein; DSS, dextran sulfate sodium; SAA, Serum amyloid antigen; TNBS, trinitrobenzenesulfonic acid.

## Data Availability

The data presented in this study are available on request from the corresponding author.

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
