# Peer review of "Response Variability to Drug Testing in Two Models of Chemically Induced Colitis"

_ijms, 2023, doi:10.3390/ijms24076424_

Round 1

Reviewer 1 Report

This work is well written in general and presents a detailed approach to methods. The sex-based differences of ROS assessed by bioluminescence turn out to be interesting. This element is raised by the authors as to be considered when performing experiments on drug screening by TNBS- and DSS-induced colitis. There are several sections from the manuscript (see below) that are lacking in bibliographical references. 

- Lines 38-39: "by a relapse and remission pattern," is not very clear, perhaps it would be better to say "relapsing-remitting pattern" to mean a course characterized by episodes of remission and relapse of disease;

- Lines 36-43: These are all agreeable sentences but lack bibliographical notes. It would be correct to include them;

- Lines 48-51: Bibliographical notes should also be put here;

- Lines 58-63: bibliographical notes need to be put here too;

- Lines 65-72: bibliographical notes need to be put here as well;

- In Figure 1,2 The tables (C) are barely legible as are the legends of the other graphs: perhaps the image should be enlarged a bit or its resolution improved?

- Lines 297-298: a bibliographical note (indeed more) should be put here citing the "few studies" the authors refer to.

Author Response

Dear Reviewer,

We are grateful for the comments we have received and we are sure these will definitely  improve the quality and impact of our work. We hereby respond to the comments and suggestions the reviewer has made to improve our paper. First of all, we have applied all the suggestions we received and extensively reviewed the bibliography to better reference the article:

  • Lines 38-39: "by a relapse and remission pattern," is not very clear, perhaps it would be better to say "relapsing-remitting pattern" to mean a course characterized by episodes of remission and relapse of disease;
    • We have changed this expression to the suggested one (lines 38-39).
  • Lines 36-43: These are all agreeable sentences but lack bibliographical notes. It would be correct to include them;
    • We have added appropriate references to describe both Crohn’s disease and Ulcerative colitis and their etiology (Now lines 38-46).
  • Lines 48-51: Bibliographical notes should also be put here;
    • References describing the different colitis models and their inability to reproduce the human disease have been added (now lines 48-50).
  • Lines 58-63: bibliographical notes need to be put here too;
  • We have put references describing both TNBS and DSS models (now lines 58-64)
  • Lines 65-72: bibliographical notes need to be put here as well;
    • References describing tight junctions and cytokine profiles in colitis mice have been included (now lines 68-71).
  • In Figure 1,2 The tables (C) are barely legible as are the legends of the other graphs: perhaps the image should be enlarged a bit or its resolution improved?
    • Both images were enlarged in the manuscript and tables were made bigger (font and whole table). Resolution was already high (300dpi) so we needed to enlarge the tables and whole figure.
  • Lines 297-298: a bibliographical note (indeed more) should be put here citing the "few studies" the authors refer to.
    • We have rephrased this line as there are many studies using both models, we have cited some representative ones (now lines 301- 303).

We hope this responds to the reviewer comments and suggestions.

Kind regards,

Josep Manyé.

Reviewer 2 Report

Title:

Response variability to drug testing in two models of chemically induced colitis

Summary:

Author has performed an interesting research study depicting the use of TNBS and DSS induced colitis model to evaluate the pathology of IBD and drug screening. With preventive and therapeutic experimental trial, author has also elucidated the drug screening variability and inflammatory response in male and female mice. Author suggested that immunity of each model, the course of inflammation and sex must be considered into experimental design. However, there are some minor corrections that needs to rectify in the manuscript.

Suggestions:

1-     Author needs to add full form of every abbreviation used first time

2-     Introduction is under referenced; author is suggested to include more studies specific to the problem

3-     Author is suggested to point out other studies performed on the sex stratification in chemically induced colitis and their findings

4-     Since author has taken ROS as the most distinguished marker for inflammation, it is suggested to the author to incorporate the ROS estimation method in method section and also give stress to the implication and importance of ROS estimation in introduction section also

5-     In line 349 – “Specific inflammatory or reparation markers are therefore required to validate this in vivo imaging method”

Author is suggested to elaborate this sentence envisaging specific inflammatory and reparation makers required to validate in vivo imaging method  

1-     The author used TNBS and DSS induced colitis models to understand the response mechanism and variability in male and female mice. Author is suggested to envisage which gender based model is suitable for what type of studies, if possible. It will elaborate the novelty of the research study.

Author Response

Dear Reviewer,

We acknowledge the interesting and proper comments received and we are sure these have improved the quality and relevance of our work. We hereby respond to the comments and suggestions the reviewer have made to improve our paper:

  • Author needs to add full form of every abbreviation used first time;
    • List of abbreviations was added before references (line 646).
  • Introduction is under referenced; author is suggested to include more studies specific to the problem;
    • Introduction was appropriately referenced after revision: for a proper description of both Crohn’s disease and Ulcerative colitis and their etiology (lines 38-46); of the different colitis models and their inability to reproduce the human disease have been added (lines 48-50); and of both TNBS and DSS models (lines 58-64).
  • Author is suggested to point out other studies performed on the sex stratification in chemically induced colitis and their findings;
    • We have pointed out some studies focused on the differences between sexes in TNBS and DSS colitis (lines 75-81 and 392-395).
  • Since author has taken ROS as the most distinguished marker for inflammation, it is suggested to the author to incorporate the ROS estimation method in method section and also give stress to the implication and importance of ROS estimation in introduction section also;
    • We have clarified how the ROS quantification is done by using the bioluminescent probe L-012, which produces a chemiluminescent signal in the presence of ROS (lines 543-545). Morevoer, we have added to the introduction a section explaining the importance of ROS in IBD and the use of this bioluminescent technique in chemically induced colitis models in order to evaluate the inflammatory course (now lines 85-102). We have also modified a line in the discussion to link it to the introduction new section (now line 319-322).
  • In line 349 – “Specific inflammatory or reparation markers are therefore required to validate this in vivo imaging method”. Author is suggested to elaborate this sentence envisaging specific inflammatory and reparation makers required to validate in vivo imaging method;
    • Now line 350. We have build a table (Table 4) to propose inflammatory, reparation or oxidative stress markers for each of the models, approach (therapeutic or preventive) and sex to validate ROS detection using the bioluminescence in our experimental conditions.
  • The author used TNBS and DSS induced colitis models to understand the response mechanism and variability in male and female mice. Author is suggested to envisage which gender based model is suitable for what type of studies, if possible. It will elaborate the novelty of the research study.
    • As far as we are concerned, studies should use both sexes but understanding the potential differences that will appear in colitis induction, the inflammatory course and drug efficacy between sexes. Furthermore, at the same time, working in both sexes helps us build the mechanism of action with more accuracy. Appropriate designs on drug doses and administration timepoints are important and might be sex dependent.

We hope this responds to the reviewer comments and suggestions.

Kind regards,

Josep Manyé.

Round 2

Reviewer 1 Report

the revisions are appropriate.